# Outcomes of Microhook ab Interno Trabeculotomy in Consecutive 36 Eyes with Uveitic Glaucoma

**DOI:** 10.3390/jcm11133768

**Published:** 2022-06-29

**Authors:** Noriyuki Sotani, Sentaro Kusuhara, Wataru Matsumiya, Mina Okuda, Sotaro Mori, Rei Sotani, Kyung Woo Kim, Ryuto Nishisho, Makoto Nakamura

**Affiliations:** Division of Ophthalmology, Department of Surgery, Kobe University Graduate School of Medicine, 7-5-1 Kusunoki-cho, Chuo-ku, Kobe 650-0017, Japan; sn1117@med.kobe-u.ac.jp (N.S.); ytkmatsu@med.kobe-u.ac.jp (W.M.); mokuda@med.kobe-u.ac.jp (M.O.); smori@med.kobe-u.ac.jp (S.M.); rei8@med.kobe-u.ac.jp (R.S.); kim6048@med.kobe-u.ac.jp (K.W.K.); nryuto0810@gmail.com (R.N.); manakamu@med.kobe-u.ac.jp (M.N.)

**Keywords:** uveitic glaucoma, minimally invasive glaucoma surgery, microhook, intraocular pressure, complication

## Abstract

Microhook trabeculotomy (μLOT), recently developed by Tanito belongs to minimally invasive glaucoma surgery and contributes to intraocular pressure (IOP) control in eyes with glaucoma resistant to medical therapy. In this study, we aimed to investigate the effectiveness and safety of μLOT for uveitic glaucoma. The medical records of consecutive 36 eyes from 30 patients who underwent μLOT and were followed up over post-operative 1 year were reviewed. The surgical success (IOP = 5–20 mmHg and ΔIOP ≥ 20% with additional anti-glaucoma drugs) was achieved in 67% of eyes at post-operative 12 months. The median IOP significantly decreased from 30.5 mmHg pre-operatively to 15 mmHg at 12 months post-operatively (*p* = 0.001), and the median glaucoma drug score changed from 5 pre-operatively to 2.5 at 12 months post-operatively (*p* = 0.301). Intraocular inflammation scores at post-operative 6 weeks did not show a significant worsening as compared to pre-operatively, and 8 (22%) eyes exhibited exacerbation of inflammation during the 12-month follow-up period. Post-operative complications were confirmed in 58% of eyes, but most of them were mild and transient or successfully managed. With its favorable benefit–risk profile, μLOT would be an option worth considering as the first glaucoma surgery for uveitic glaucoma.

## 1. Introduction

Uveitic glaucoma is one of the chief concerns in eyes with uveitis because it can potentially lead to blindness unless properly managed. In the Systemic Immunosuppressive Therapy for Eye Diseases cohort study consisting only of cases of noninfectious inflammatory eye diseases, 13.3% of eyes were reported to have ocular hypertension (≥21 mmHg) at cohort entry [1]. It is well known that the incidence and prevalence of uveitis differ by geographic location [2], but previous reports from different countries showed the incidence of uveitic glaucoma ranging from 6.5% to 41.8% [3,4,5,6,7,8], suggesting that uveitic glaucoma is a common problem all over the world. In a retrospective study analyzing 1076 patients with uveitis between 2011 and 2013, secondary glaucoma, along with chronic cystoid macular edema, macular scarring, and epiretinal membrane, is a cause of moderate or severe vision loss [9]. With the advent of tumor necrosis factor-alpha inhibitors and the advances in the surgical system, ocular complications other than secondary glaucoma could be better managed in recent years [10,11,12,13,14]. Accordingly, uveitic glaucoma is becoming more important as a major cause of vision loss.

The management of uveitic glaucoma consists of controlling the intraocular inflammation and reducing intraocular pressure (IOP). Regarding IOP control in eyes with uveitis, glaucoma surgery should be timely performed if medical therapy fails to control IOP because eyes with uveitis can rapidly progress from ocular hypertension to uveitic glaucoma [15] and visual field loss occurs faster in glaucoma patients with uveitis than those without uveitis [16]. Since there is no evidence-based guideline for uveitic glaucoma, it is left to the discretion of the physician on how to treat individual patients. Probably, most uveitis specialists would follow the guidelines for primary glaucoma [17]. For instance, in open-angle eyes with uveitis, the initial target IOP might be set as 75% (25% reduction) of the pretreatment IOP, and medical and/or surgical treatment is to be selected to achieve the target IOP.

Surgical treatment for open-angle glaucoma has changed dramatically with the development of minimally invasive glaucoma surgery (MIGS). MIGS is a generic term for a new class of glaucoma drainage devices and procedures that is safer and less invasive than conventional glaucoma surgeries [18,19,20], and therefore would be a more suitable treatment option for uveitic glaucoma in which surgical stress can cause severe ocular inflammation, post-operative scarring, or both. Since 2015, microhook trabeculotomy (μLOT), an ab interno trabeculotomy using metal microhooks devised by Tanito M [21,22] and classified as MIGS, has been performed in glaucoma surgery mainly in Japan. In this study, we investigated the effectiveness and safety profile of μLOT in eyes with uveitic glaucoma.

## 2. Materials and Methods

### 2.1. Subjects

In this single-center retrospective study, we reviewed the medical records of consecutive patients who had been treated with μLOT for uveitic glaucoma (including ocular hypertension in eyes with uveitis) during the period from February 2017 to August 2019 at Kobe University Hospital and were followed up over 1 year after surgery. This study did not set exclusion criteria because it aimed to assess the clinical outcome of μLOT for uveal glaucoma in a real-world clinical setting. Approval for this study was granted by the institutional review board of the Kobe University Graduate School of Medicine (permission number: B200091). This study adhered to the tenets of the Declaration of Helsinki for research on human subjects, and the IRB exempted us from obtaining informed consent from the patients due to the retrospective and observational nature of this study. However, the patients were given the opportunity to express the choice for the data to be used using an opt-out system through the hospital website.

### 2.2. Data Collection

The collected data were as follows: age, sex, operated eye, anatomical type of uveitis, cause of uveitis, previous intraocular surgery, lens status, decimal best-corrected visual acuity (BCVA) (converted to the logarithm of the minimum angle of resolution [logMAR] for analyses), intraocular pressure measured using Goldmann applanation tonometer, glaucoma drug score, anterior chamber cell, anterior chamber flare, vitreous haze, the presence/absence of retinal/choroidal inflammatory lesion, corneal endothelial cell density (CECD), mean deviation of visual field calculated by Humphrey field analyzer 24-2/30-2 visual field test, use of anti-inflammatory and anti-thrombotic drugs, concomitant surgery, incision range of trabecular meshwork, and surgical complications. Intraocular inflammation for anterior chamber cell, anterior chamber flare, and vitreous haze was scored according to the National Eye Institute criteria adapted by the Standardization of Uveitis Nomenclature Working Group [23].

### 2.3. Surgical Procedure

μLOT was performed according to an original procedure devised by Tanito et al. [21] with a minor modification as previously described [24,25]. In brief, for eyes undergoing 1-quadrant μLOT, after a corneal paracentesis was created, the aqueous humor was replaced by cohesive viscoelastic. Then, the patient’s head was rotated away from the surgeon and the microscope was tilted towards the surgeon to obtain a clear view of the angle through a Swan-Jacob gonioprism lens. After insertion of a straight Tanito microhook (M-2215 s, Inami & Co., Ltd., Tokyo, Japan) into the anterior chamber, the trabecular meshwork was incised over a 120° nasal area (Figure 1) (see Appendix A). Viscoelastic was then irrigated from the anterior chamber and the corneal wound was hydrated to ensure water-tight closure. In case of 2-quadrant μLOT, an additional incision of the trabecular meshwork was performed over a 120° temporal area through a nasal corneal port. Post-operatively, eye drops (antibiotic, steroid, 2% pilocarpine, and non-steroidal anti-inflammatory drug [in case of combined cataract surgery]) were administered approximately for 1 month.

### 2.4. Outcomes

The primary outcome measure was surgical success rate at post-operative 12 months, and secondary outcomes were changes in IOP and glaucoma drug score preoperatively to 12 months, changes in intraocular inflammation scores preoperatively to 6 weeks, and surgical complications. The surgical failure was defined as any of the following: (1) IOP of less than 5 mmHg or more than 20 mmHg on two consecutive visits, (2) IOP reduction of less than 20% from preoperative IOP on two consecutive visits, and (3) reoperation. Surgical success was defined as the lack of all above criteria, without antiglaucoma medication (complete success) or with antiglaucoma medications (qualified success) after post-operative 4 weeks. Glaucoma drug score (GDS) was the number of anti-glaucoma eye drops. Two points were given for the combination of two types of anti-glaucoma drugs or for oral carbonic anhydrase inhibitor. Intraocular inflammation was evaluated by the standardized scoring system mentioned above and fundoscopic findings. In assessing surgical complications, hyphema was counted if layered accumulation of red blood cells within the anterior chamber was observed. Post-operative IOP spike was defined as transient IOP elevation exceeding 30 mmHg within post-operative 2 weeks.

### 2.5. Statistical Analyses

In statistical analyses, the data of IOP and GDS after additional glaucoma surgery were handled as missing data, and the missing data were imputed using the last observation carried forward method. A liner-mixed model was used for comparison of variables at two different time points (pre-operatively/post-operative 6 weeks or preoperatively/post-operative 12 months). Kaplan–Meier analysis was performed to evaluate success rate of surgery. Statistical analyses were performed using MedCalc v.20.027 software (MedCalc Software, Ostend, Belgium) or EZR (Saitama Medical Center, Jichi Medical University, Saitama, Japan), which is a graphical user interface for R software (The R Foundation for Statistical Computing, Vienna, Austria). A *p*-value less than 0.05 was considered significant.

## 3. Results

### 3.1. Preoperative Characteristics

A total of 36 eyes from 30 patients were included in this study. The summary of preoperative characteristics is shown in Table 1. 

The median age was 68.5 years, with females predominant, and about one-third of uveitis was caused by sarcoidosis. All eyes had an open angle, and the median (interquartile range [IQR]) of CECD was 2584 (2237, 2778) cells/mm^2^. The median (IQR) of decimal BCVA was 1.0 (0.5, 1.2). Of 36 eyes, 21 (58%) eyes had a history of intraocular surgery: cataract surgery (19 [53%] eyes), vitreous surgery (8[22%] eyes), and glaucoma surgery (trabeculectomy, ab externo trabeculotomy, or ab interno trabeculotomy) (5 [14%] eyes) (there is some overlapping). The median (IQR) score of anterior chamber cell, anterior chamber flare, vitreous haze was 0.0 (0.0, 0.0), 0.0 (0.0, 0.0), and 0.0 (0.0, 0.0), respectively, and retinal and/or choroidal inflammatory lesion was present in 2 (6%) eyes. Corticosteroid was administered topically in 24 (67%) eyes and systemically in 2 (7%) patients. The use of antithrombotic drugs was confirmed in 4 (11%) eyes.

### 3.2. Surgical Outcomes

Of 36 eyes, 27 (75%) eyes were treated with singly surgery (μLOT) and 9 (25%) eyes with combined surgery (μLOT + cataract surgery). The incision quadrant was 1 quadrant in 24 (67%) eyes and 2 quadrants in the remaining 12 (33%) eyes. Surgical success rates at post-operative 12 months were shown as survival probability in Figure 2. 

The probability of complete success was 19% while that of qualified success was 67%. Figure 2 clearly demonstrates that most surgical failures were recognized within 1 month after surgery. The median (IQR) IOP decreased as compared to 30.5 (24.75, 39) mmHg pre-operatively, 16.5 (12.75, 20.25) mmHg at 1 month post-operatively, 14.5 (11, 20.25) mmHg at 3 months post-operatively, 14 (12, 22.25) mmHg at 6 months post-operatively, 15 (11, 22.5) mmHg at 9 months post-operatively, and 15 (11, 23.25) mmHg at 12 months post-operatively (*p* = 0.001 between pre-operatively and 12 months post-operatively) (Figure 3). 

The median (IQR) GDS also decreased after μLOT from 5 (4, 6) pre-operatively to 2 (0, 3) at 1 month post-operatively, 2 (0, 4.25) at 3 months post-operatively, 2 (0, 4.25) at 6 months post-operatively, 2 (0.75, 4.25) at 9 months post-operatively, and 2.5 (1. 4.25) at 12 months post-operatively. However, the degree of the decrease did not reach a significant level (*p* = 0.301 between pre-operatively and 12 months post-operatively) (Figure 4).

### 3.3. Change in Inflammation

Intraocular inflammation at post-operative 6 weeks did not show a significant worsening as compared to pre-operatively. The median (IQR) score of anterior chamber cell, anterior chamber flare, vitreous haze at 6 weeks post-operatively was 0.0 (0.0) (*p* = 0.536), 0.0 (0.0) (*p*-value was not obtained), and 0.0 (0.0) (*p* = 0.717), respectively, and the presence of retinal and/or choroidal inflammatory lesion was confirmed in 4 (11%) eyes. However, 8 (22%) eyes were rated as having an exacerbation of inflammation during the 12-month follow-up period.

### 3.4. Complications

Post-operative complications were observed in 21 (58%) eyes. The most frequent complication was IOP spike (33%), followed by worsening of inflammation (22%), hyphema (14%), macular hole formation (6%), transient hypotony (3%), cataract progression (3%), and ocular pain (3%) (there were cases with multiple complications). Additional glaucoma surgery was needed in nine (25%) eyes: trabeculectomy in seven eyes, Ahmed glaucoma valve in one eye, and Baerveldt glaucoma implant in one eye, with the median (IQR) time to additional surgery of 4.0 (2.5, 8.0) months. The median (IQR) time of worsening of inflammation after surgery was 2.0 (1.8, 8.0) months. The median (IQR) logMAR BCVA did not show a significant change from 0.000 (−0.079, 0.301) pre-operatively to 0.000 (−0.176, 0.222) at 12 months post-operatively (*p* = 0.829), and the median (IQR) CECD of 2551.5 (2148, 2819) cells/mm^2^ at 1 month was not significantly different from the pre-operative value (*p* = 0.343). 

## 4. Discussion

The best strategy for the surgical management of uveitic glaucoma remains controversial. However, it seems reasonable to start with less invasive surgery if situations permit. Therefore, ab interno incision of the trabecular meshwork using a Tanito microhook through the small corneal port (μLOT) may be a preferred method as the first glaucoma surgery for eyes with uveitic glaucoma.

Preoperative characteristics in our study are roughly comparable to but slightly different from those in previous studies in which uveitic glaucoma was treated with non-filtering surgery: viscocanalostomy, trabectome, ab externo trabeculotomy, or Kahook Dual Blade [26,27,28,29]. The number of eyes (*n* = 36) and the median age (68.5 years) in our study were higher than those in the previous studies (*n* = 11–24; 38–52 years). Females accounted for 61% of our study, which is a similar percentage to the previous studies (58–63%) except for one study by Voykov et al. (18%) [26,27,28,29]. The most frequent cause of uveitis was different among studies: sarcoidosis in our study, sarcoidosis, Fuchs’ heterochromic iridocyclitis, and pars planitis in the previous studies [26,27,29]. The median or mean IOP was similar among studies: 30.5 mmHg in our study and 27–35.6 mmHg in the previous studies, while the median or mean GDS was higher in our study as compared to the previous studies (5 vs. 2–5) [26,27,28,29]. 

Overall, the surgical success rate in our study was promising because two-thirds of eyes had a favorable IOP control (IOP = 5–20 mmHg and ΔIOP ≥ 20%) with the aid of anti-glaucoma eye drops (qualified success) at post-operative 12 months. Although it should be noted that the definition of surgical success differs among studies, the surgical success rate in our study seems to be non-inferior to that in previous studies in which eyes with quiescent uveitic glaucoma underwent non-filtering surgery or eyes with all types of glaucoma were treated with μLOT. Miserocchi et al. assessed 11 patients with uveitic glaucoma who had been treated with viscocanalostomy and had been followed up for 23 to 56 months after surgery. In this study, 90.9% achieved qualified success (IOP = 6–21 mmHg with 1 or more anti-glaucoma medications, goniopuncture, or both) [26]. Anton et al. treated 24 patients with uveitic glaucoma by Trabectome and followed the patients for 60–1046 days. Although none of the patients achieved absolute success (IOP < 21 mmHg and ΔIOP ≥ 20% without additional anti-glaucoma medications), 87.5% attained relative success (IOP < 21 mmHg and ΔIOP ≥ 20% with additional anti-glaucoma medications) (the information on time point was not provided) [27]. Voykov et al. summarized the surgical outcomes of conventional ab externo trabeculotomy for uveitic glaucoma. Of 22 eyes, 11(50%) eyes fulfilled qualified success (IOP = 6–21 mmHg and ΔIOP ≥ 25% with additional anti-glaucoma medications) at post-operative 1 year [28]. Miller et al. evaluated the effectiveness of goniotomy with Kahook Dual Blade in 16 patients with uveitis-associated ocular hypertension or glaucoma and reported that the probability of surgical success (ΔIOP ≥ 20% with ongoing medical therapy for ocular hypertension and ΔIOP ≥ 30% with ongoing medical therapy for glaucoma) was 68% at post-operative 1 year [29]. Regarding the surgical success after μLOT for all types of glaucoma, Tanito et al. carried out a single-center retrospective study and analyzed 560 cases in which primary open-angle glaucoma and exfoliation glaucoma accounted for 57% and 20%, respectively. The surgical success rate at 1 year was 44.6% (IOP ≤ 18 mmHg and ΔIOP ≥ 20% with the use of anti-glaucoma medications) and 69.1% (IOP ≤ 18 mmHg and ΔIOP ≥ 0% with the use of anti-glaucoma medications) [30]. Recently, Mori et al. reported the results of a multicenter retrospective study in which 392 patients with all types of glaucoma were treated with μLOT. Surgical success in that study (IOP = 5–21 mmHg, ΔIOP ≥ 20%, and no additional glaucoma surgery) was 74.2% at 1 year after surgery [31].

In glaucoma surgery, an IOP decrease is an important outcome next to surgical success. The median IOP of 15 mmHg 12 months after surgery in our study was favorable because the target IOP in uveitic glaucoma is usually set to below 20 mmHg unless advanced glaucoma. The reported median or mean IOP after non-filtering glaucoma surgery for uveitic glaucoma ranges from 15 mmHg to 18.1 mmHg [26,27,28,29], which is slightly higher than in our study. Regarding GDS, the median value of 2.5 in our study was higher than that in the previous study (0.67–2.1) [26,27,28,29]. Together with the less median IOP value, the eyes might have been aggressively treated with anti-glaucoma medications after surgery in our cohort.

In general, much attention should be paid to post-operative intraocular inflammation in surgery for uveitic glaucoma since surgical stress is likely to cause or exacerbate ocular inflammation in eyes with uveitis. However, no standardized assessment of intraocular inflammation has been performed after non-filtering glaucoma surgery for uveitic glaucoma although some papers mentioned the presence/absence of inflammation or cystoid macular edema after surgery [26,27,28,29]. In our study, intraocular inflammation was evaluated by the standardized scoring system widely used all over the world and fundoscopic findings. At post-operative 6 weeks, no significant exacerbation of intraocular inflammation was observed in the anterior chamber and vitreous cavity, but 11% of eyes exhibited retinal and/or choroidal inflammatory lesion. In addition, exacerbation of inflammation occurred in 22% of eyes during the 12-month follow-up period. It is unclear whether the signs of post-operative inflammation were associated with μLOT-related surgical insult, but it should be noted that even MIGS also can cause post-operative inflammation in eyes with uveal glaucoma. 

Post-operative complications were confirmed in 58% of eyes in our study. The percentage is higher than that in previous studies in which non-filtering glaucoma surgery was performed for uveitic glaucoma (4–45%) [26,27,28,29] but is not so high as compared to 43% in Tanito’s study in which μLOT was performed for all types of glaucoma [30]. We attribute the higher complication rate in our study to the detailed observation and/or recordings of ocular findings after surgery. The point is that most of the complications were mild and transient or successfully managed in our study, which would contribute to a favorable benefit–risk profile of μLOT for uveitic glaucoma.

The main limitations of the current study stem from the small number of subjects and retrospective nature which should be overcome in a large-scale prospective study in the future. Short follow-up time is another limitation. Because eyes with uveitis usually experience relapses of ocular inflammation and long-term use of corticosteroids, both of which affect IOP control, long-term data of μLOT for uveitic glaucoma should be collected and analyzed in the future. The difference in incision quadrant might affect the surgical outcome in μLOT for uveitic glaucoma. Mori et al. compared the effectiveness and safety of μLOT for all types of glaucoma between 1-quadrant and 2-quadrant incision groups and concluded that the 1-year success rate was not significantly different, but 2-quadrant incision groups showed a significantly higher rate of transient IOP rise post-operatively [24]. Further, the relationship between the prolonged use of anti-glaucoma medications and surgical outcomes should be investigated in the future because Okuda et al. recently reported that the prolonged use of anti-glaucoma drugs was significantly associated with surgical failure after μLOT for all types of glaucoma [25].

In conclusion, in μLOT for uveitic glaucoma, good IOP control can be expected in about 2/3 of cases 1 year after surgery with additional anti-glaucoma medications. Although post-operative complications would occur in about 50% of cases, most of the complications would be mild and transient or successfully managed. With its favorable benefit–risk profile, μLOT would be an option worth considering as the first glaucoma surgery for uveitic glaucoma.

## Figures and Tables

**Figure 1 jcm-11-03768-f001:**
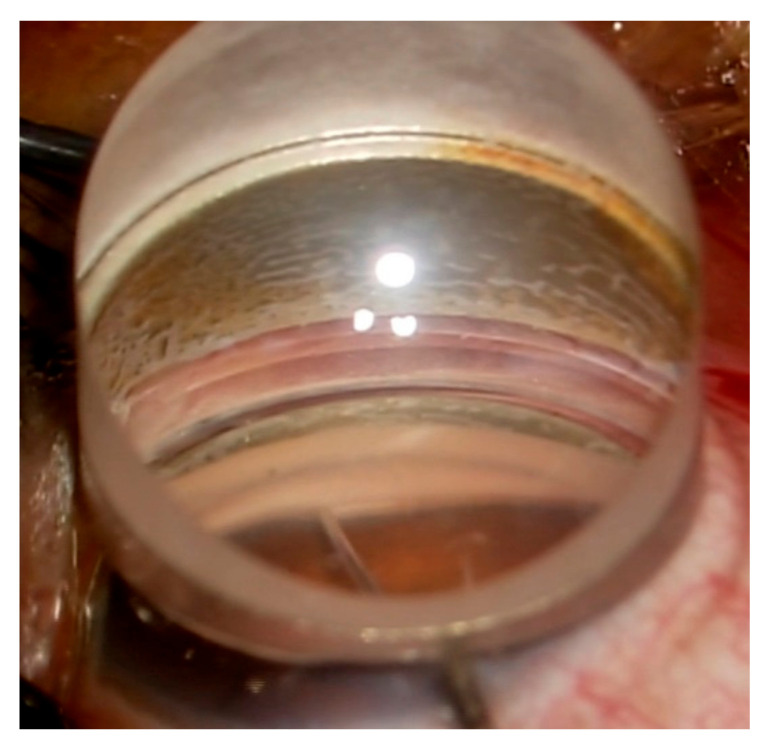
Intraocular picture of microhook trabeculotomy. The inner wall of the Schlemm’s canal and trabecular meshwork is being incised using Tanito microhook.

**Figure 2 jcm-11-03768-f002:**
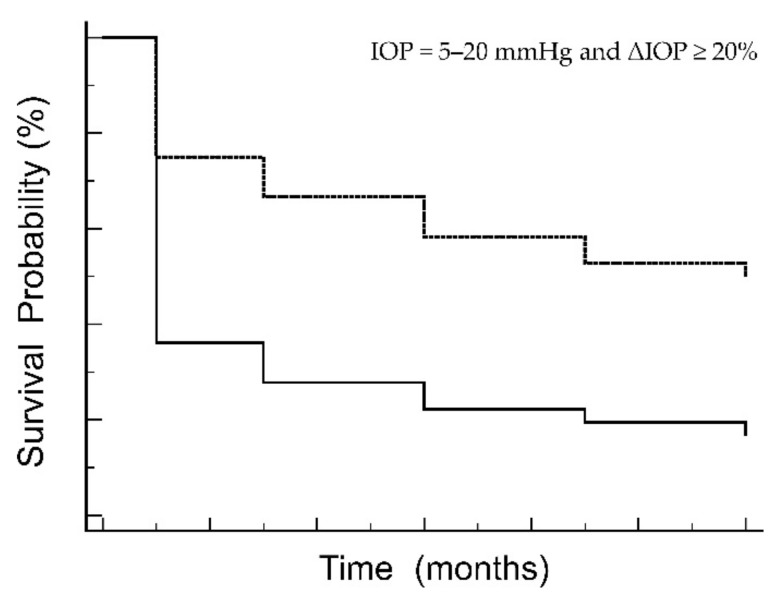
Kaplan–Meier survival curves for surgical success. Continuous line: complete success (IOP = 5–20 mmHg and ΔIOP ≥ 20% without additional anti-glaucoma medications). Broken line: qualified success (IOP = 5–20 mmHg and ΔIOP ≥ 20% with additional anti-glaucoma medications). IOP = intraocular pressure.

**Figure 3 jcm-11-03768-f003:**
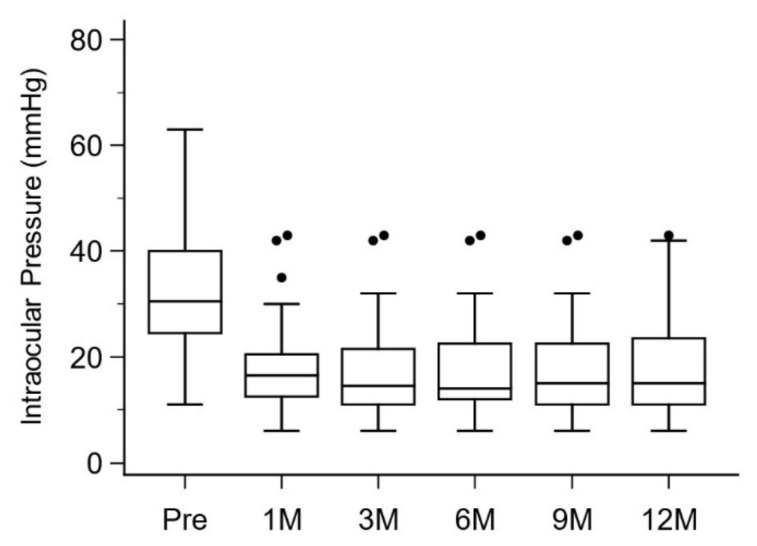
Box and whisker plots of intraocular pressure at different time points. Pre, pre-operatively; 1 M, 1 month post-operatively; 3 M, 3 months post-operatively; 6 M, 6 months post-operatively; 9 M, 9 months post-operatively; 12 M, 12 months post-operatively.

**Figure 4 jcm-11-03768-f004:**
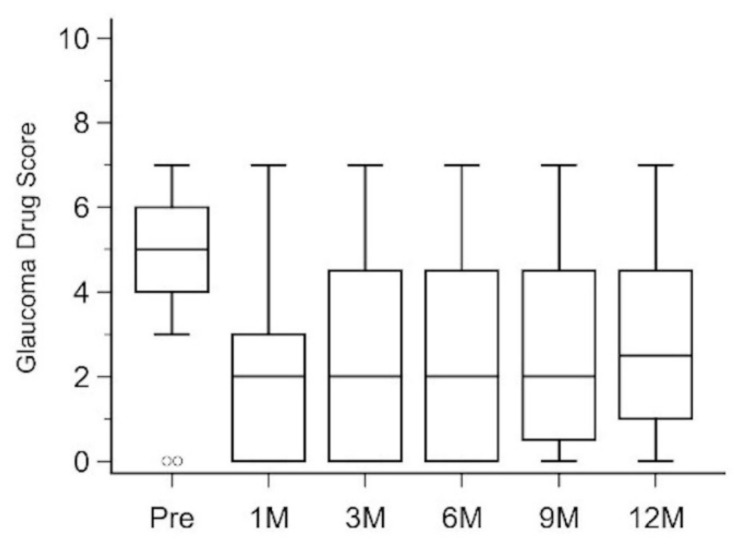
Box and whisker plots of glaucoma drug score at different time points. Pre, pre-operatively; 1 M, 1 month post-operatively; 3 M, 3 months post-operatively; 6 M, 6 months post-operatively; 9 M, 9 months post-operatively; 12 M, 12 months post-operatively.

**Table 1 jcm-11-03768-t001:** Patient Preoperative Characteristics.

Characteristics		Data
Number of patients/affected eyes, *n*/*n*		30/36
Age (years), median (IQR)		68.5 (52.5, 72)
Sex, *n* (%)		
	Male	11 (37)
	Female	19 (63)
Eye, *n* (%)		
	Right	20 (56)
	Left	16 (44)
Cause of uveitis, *n* (%)		
	Sarcoidosis	11 (31)
	Posner–Schlossman syndrome	7 (19)
	Cytomegalovirus anterior uveitis	4 (11)
	Vogt–Koyanagi–Harada disease	3 (8)
	Scleritis	3 (8)
	Syphilitic uveitis	1 (3)
	Behçet’s disease	1 (3)
	Unclassified	6 (17)
Previous intraocular surgery, *n* (%)		
	Cataract surgery	19 (53)
	Vitreous surgery	8 (22)
	Glaucoma surgery	5 (14)
Best-corrected visual acuity (decimal), median (IQR)		1.0 (0.5, 1.2)
Best-corrected visual acuity (logMAR), median (IQR)		0.000 (−0.079, 0.301)
Intraocular pressure (mmHg), median (IQR)		30.5 (24.75, 39)
Glaucoma drug score, median (IQR)		5 (4, 6)
Mean deviation of visual field (dB), median (IQR)		−11.9 (−17.0, −5.36)
Use of antithrombotic drugs, *n* (%)		
	yes	4 (11)
	no	32 (89)

Abbreviations: IQR, interquartile range; logMAR, logarithm of the minimum angle of resolution; dB, decibels.

## Data Availability

Not applicable.

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
