# Peer review of "Outcomes of Microhook ab Interno Trabeculotomy in Consecutive 36 Eyes with Uveitic Glaucoma"

_jcm, 2022, doi:10.3390/jcm11133768_

Round 1
Reviewer 1 Report
It is an interesting study to expand sugical techniques in glaucoma surgery.
It is recommanded to add or explain more information as below.
1. Exclusion criteria are supposed to be described to understand characteristics of uveitic patients who underwent uLOT.
2. It is recommanded to analyse risk factors associated with surgical failure using statistical method like Cox proportional hazard model. Preoperative factors like age, gender, preoperative IOP, types of previous surgery should be included for the analysis. Preoperative VF MD may represent a surrogate for angle degeneration distal to TM and might be an important factor for surgical success because ab interno trabeculotomy is an angle surgery. Other factors like the number of quadrants involved during uLOT procedure could be included for the analysis.
3. Previous surgical procedures are recommanded to be described in detail. Macular hole formation is a rare complication after glaucoma surgery. Is it related to recurrence of uveitis or previous vitreoretianl procedures? Five eyes underwent glaucoma surgery were included in this study. What are the names of glaucoma surgical procedures?
4. How many eyes showed IOP spike did not respond to medical therapy and needed additional surgery?
5. Time interval between initial surgey and second procedure like Trab or GDD implantation need to be described (median and extent of the days or months).
Reviewer 2 Report
The study titled " Outcomes of microhook ab interno trabeculoltomy in consecutive 36 eyes with uveitic glaucoma” investigates the effectiveness and safety of microhook trabeculotomy for uveitic glaucoma. The manuscript is well-written and clear. Nevertheless, there are still concerns, questions, and comments that are detailed below.
[Introduction]
Page 2, line 55-56: the guideline for primary open angle glaucoma
à It would be better to edit it in the bibliography format instead of the website address.
[Materials and Methods]
Page 2, line 88: glaucoma drug score
à It is explained in more detail in ‘Outcomes’ section, but it would be better to define what it means in this section.
Why did you investigate taking anti-thrombotic drugs? If taking it, did the patient stop taking it before surgery? Did taking the drug affect the surgical outcome or complications?
One of the causes of increased IOP in uveitic glaucoma is the presence of PAS. Did you evaluate the angle status before surgery?
What was the indication for the 2-quadrant mLOT?
Page 3, line 120-121
à Why was it given 2 points when taking Oral carbonic anhydrase inhibitor?
[Discussions]
Because of inflammation in uveitic glaucoma, it is known that the outcomes of conventional surgery (glaucoma filtering surgery or drainage devices) are not as good as those of other glaucoma. What are the advantages of mLOT for uveitic glaucoma?
Author Response
Thank you for your time and efforts to review our manuscript while you are busy. Our responses to your comments are itemized below.
- [Introduction] Page 2, line 55-56: the guideline for primary open angle glaucoma à It would be better to edit it in the bibliography format instead of the website address.
According to your advice, we moved the website address to the reference section in our revised manuscript, which is subject to the journal’s guideline.
- [Materials and Methods] Page 2, line 88: glaucoma drug score à It is explained in more detail in ‘Outcomes’ section, but it would be better to define what it means in this section.
‘Outcomes section’ is a subsection of Materials and Methods section, and GDS is one of the important outcomes in this study. Therefore, we put the definition of GDS in ‘Outcomes section.’ We think that the way and the degree of explanation of GDS in this manuscript are similar to those in previous reports.
- Why did you investigate taking anti-thrombotic drugs? If taking it, did the patient stop taking it before surgery? Did taking the drug affect the surgical outcome or complications?
As you pointed out, the effect of anti-thrombotic drugs in surgical results would be of great interest for most of glaucoma specialist because it may increase bleeding events intra- and perioperatively. It was at the discretion of the physician whether stopping anti-thrombotic drug before and after μLOT. Unfortunately, only 4 eyes took anti-thrombotic drug, meaning that the sample size was not enough to statistically evaluate the impact of drug intake on surgical outcomes.
- One of the causes of increased IOP in uveitic glaucoma is the presence of PAS. Did you evaluate the angle status before surgery?
As described in Results section, all eyes had an open-angle. Although some low-height and partial PAS may have existed in some cases, its impact on IOP and/or surgical outcomes would be limited. However, we agree to your idea and will plan a clinical study focusing on preoperative angle status in the future.
- What was the indication for the 2-quadrant mLOT?
It was at surgeon’s discretion.
- Page 3, line 120-121 à Why was it given 2 points when taking Oral carbonic anhydrase inhibitor?
Oral carbonic anhydrase inhibitor has greater impact on IOP decrease than eye drop. So, we gave 2 points. This scoring system is popular and has been used in many clinical studies so far.
- [Discussions] Because of inflammation in uveitic glaucoma, it is known that the outcomes of conventional surgery (glaucoma filtering surgery or drainage devices) are not as good as those of other glaucoma. What are the advantages of mLOT for uveitic glaucoma?
“What are the advantages of μLOT for uveitic glaucoma” is exactly why we planned and carried out this study. As stated in conclusion paragraph, one of the advantages of μLOT for uveitic glaucoma is its favorable benefit risk profile.

Reviewer 3 Report
please add some pictures and a video of the surgery
Author Response
Thank you for your time and efforts to review our manuscript while you are busy. Our response to your comment is as follows:
- please add some pictures and a video of the surgery
Thank you for your constructive opinion. We added an intraocular picture as Figure 1 and provided a surgical video as Supplemental Material.
